# Energy Requirements and Nutritional Strategies for Male Soccer Players: A Review and Suggestions for Practice

**DOI:** 10.3390/nu14030657

**Published:** 2022-02-04

**Authors:** Andrew T. Hulton, James J. Malone, Neil D. Clarke, Don P. M. MacLaren

**Affiliations:** 1Department of Nutritional Sciences, Faculty of Health and Medical Sciences, University of Surrey, Guildford GU2 7XH, UK; 2School of Health and Sport Sciences, Liverpool Hope University, Liverpool L16 9JD, UK; malonej2@hope.ac.uk; 3Centre for Sport, Exercise and Life Sciences, Coventry University, Coventry CV1 5FB, UK; ab1633@coventry.ac.uk; 4Research Institute for Sport and Exercise Sciences, Liverpool John Moores University, Liverpool L3 3AF, UK; dpmmaclaren@btinternet.com

**Keywords:** soccer, energy intake, energy expenditure, carbohydrate, protein, hydration, match day nutrition

## Abstract

Soccer is a high intensity intermittent sport, featuring critical events completed at high/maximal intensity which is superimposed onto an aerobic base of lower intensity activities and rest. Due to these varying energic demands and the duration of competition the need for optimal nutritional strategies to offset and delay fatigue are paramount. Over the last 50 years, several investigations have been reported on aspects of soccer be they nutrition-focused or those concerning the demands of the sport. Emanating from these scientific papers, observations have been made on the likely factors which result in the fatigue during match-play. Factors such as muscle glycogen depletion and hypoglycaemia are discussed. Studies on the energy demands of soccer have employed a variety of methodologies which are briefly reviewed and vary between the use of heart rate telemetry to the use of global positioning systems (GPS). Moving on from observations of the energy demands of the sport leads to the major focus of this review which highlights key nutritional strategies to support the preparation and recovery of male soccer players to enhance performance, or at least to enable players to perform adequately. This review examines relevant methodologies in assessing training and competitive energy costs as well as the concomitant energy intakes demanded for successful performance outcomes. In order to bring an applied aspect to the overall findings from areas discussed, some practical ideas of feeding strategies are presented.

## 1. Introduction

Soccer is a highly demanding intermittent sport that involves fluctuations between low and high intensity activities [1]. At the professional level, soccer players typically cover around 11–13 km per match dependent on position, with central midfielders covering the highest and central defenders the lowest distances [2]. Of the overall distance covered, around 1150 m is run at speeds above 20 km·h^−1^, with about 60 sprints performed; this is again dependent on the positional role [3]. Overall players perform more than 1200 unpredictable changes in activity, which also comprises of ~700 turns and 30–40 tackles and jumps [4]. Soccer players demonstrate significant reductions in the amount of both total and high intensity distance covered towards the last 15 min of match play compared to the initial 15-min period [5]. Additionally, players encounter temporary reductions in physical output following the most demanding 5-min period during match-play [5]. In training, soccer players typically cover around 3–7 km, with around 100–600 m and 50–400 m covered above 20 km·h^−1^ and 25 km·h^−1^, respectively, dependent on the training day relative to the next match (i.e., match day minus; MD-) [6,7,8]. Soccer coaches periodise the overall loading during training microcycles in order to load the player early in the training week, followed by a reduction in load as a form of taper to dissipate fatigue whilst aiming to maximise adaptation processes [9,10].

From an energy perspective, soccer places significant demand on both the aerobic and anaerobic systems in order to perform at the highest level [11]. Soccer match-play elicits average and peak heart rates of ~85 and ~98% of maximal values, respectively [12,13], and such data suggests that the average oxygen uptake is around ~70% VO2max [14]. With approximately ~90% of match-play performed at low to moderate intensity [15], the aerobic energy system provides most of the energy, with a focus being on recovery between bouts of high intensity activity [13]. High intensity activities in soccer, such as accelerations and jumping, require use of energy from the anaerobic systems [15]. These high intensity actions often lead to crucial moments of match-play, such as straight-line sprinting within goal scoring situations [16], and result in elevated levels of lactic acid which are typically 2–12 mM during soccer matches [17], although it should be noted that these values are highly influenced by the preceding five minutes of activity. Since anaerobic glycolysis as well as carbohydrate (CHO) oxidation are important sources of energy, muscle glycogen has been found to be depleted following match-play to around 43% of pre-match levels and remains significantly reduced up to and beyond 24 h hours post-match [18]. Thus, soccer players are required to be adequately prepared from both a training and nutritional perspective to meet the energy requirements.

The aim of the present review is to critically analyse the available scientific literature concerning the energy demands of both soccer training and match-play in male players, and thereafter to examine the energy intake with a view to suggesting practical guidelines. The objective is to provide evidence-based nutritional strategies for practitioners to implement in order to optimise soccer performance. Consequently, exploration of the energy expenditure as well as the metabolic changes that take place during training and matches is examined before attention is given to the nutritional needs of players.

## 2. Fatigue and Soccer

Fatigue, as defined by the inability to sustain the necessary speed or power output, in soccer has received much interest within the literature since it is important in the outcome of matches. Reilly [19] suggested the presence of fatigue could be realised in the increases in goals scored in the final 10 min of matches, although more sophisticated analysis of physical output may be determined by time-motion analysis [5] and GPS microtechnology [20]. Fatigue is seen to occur at different stages throughout a match, with temporal fatigue [5] present after intense periods of the game and attributed to disturbances in muscle ion homeostasis, to more permanent fatigue identified towards the end of match play. Nonetheless, the effect is either a temporary or more permanent inability to sustain the necessary power output. Though there are many potential mechanisms responsible, two of the main explanations which have a nutritional implication may be attributed to a depletion of muscle glycogen and hypoglycaemia [21]. Investigations from the early 1970’s [22] first demonstrated that players starting a soccer match with low muscle glycogen availability fully deplete their stores by the end of the match, although more distressing was the observation that glycogen stores were nearly depleted at half time. This results in an early onset of fatigue characterised by a significant increase in time spent at low intensity movement such as walking (50% vs. 27% of total distance), and an inability to complete sprinting activities (15% vs. 24% of total distance) compared to those who started the match with ‘normal’ glycogen availability. A comparable investigation [17] more robustly demonstrated a gradual reduction in muscle glycogen throughout a soccer match in Danish fourth division players during non-competitive matches. Although total muscle glycogen was not depleted, a reduction from 449 ± 23 mmol·kg^−1^ dry weight (dw) to 225 ± 23 mmol·kg^−1^ dw was observed and accounted for approximately 47% of muscle fibres depleted or almost depleted of glycogen. Notably, fibre type analysis was conducted and identified that depletion was apparent within type IIa and type IIx fibres, which are responsible for high intensity actions, and as such may support the significant reduction in sprinting performance observed throughout the match. It would be prudent to note that this investigation was conducted in a sub elite standard of soccer, and evidence illustrates that greater high intensity activities are performed with players of greater ability [23], therefore the prevalence of glycogen depletion may be greater than observed. Both investigations [17,22], and that of others measuring glycogen use in soccer match play [24] or simulated game activity [25,26] highlight the need for optimal glycogen storage prior to kick off for performance.

Reductions in blood glucose concentrations due to inadequate CHO availability invariably lead to hypoglycaemia since once liver glycogen is depleted there is an inability to produce sufficient glucose for circulation from the lower rates of gluconeogenesis. As the brain primarily uses glucose from the blood as an energy source, hypoglycaemia has been considered a factor contributing to fatigue. Early laboratory studies clearly demonstrated this point [27]. Consequently, hypoglycaemia may be identified as a nutritional factor that may produce fatigue or suboptimal performance in soccer [21]. The mechanism for the performance deficient due to hypoglycaemia has been attributed to affecting skilled performance and running capacity. Currell and colleagues [28] observed that the ability to dribble a ball around cones at various time points during soccer simulated activity was demonstrably slower with placebo compared with ingestion of CHO. In addition, further investigations have observed improvements in specific soccer skills [29] and perceived activation/arousal [30]. In these investigations’ hypoglycaemia was not evident, and indeed to date no studies have reported hypoglycaemia during a soccer match or in simulated studies within a laboratory context. Could it be that higher levels of blood glucose during the simulated soccer performance with glucose ingestion resulted in the greater performance of skilled based tasks and perception? Clearly more studies are required to address this issue. Current nutritional strategies are to consume CHO in the hours before and during match play, thus making the appearance of hypoglycaemia relatively rare during soccer [31]. In addition, during extended periods of play, such as extra time during tournament soccer, research has demonstrated that euglycaemia can be maintained beyond the 90 min [32].

## 3. Energy Assessment Methods in Soccer

The manipulation of energy balance within soccer players is a key nutritional consideration when attempting to achieve specific aims within the training programme. This may include the manipulation of body composition (e.g., lean body mass (LBM) or fat mass) or fuel stores (e.g., CHO periodisation) for both training and competition. The following section discusses the available methods for the assessment of energy expenditure (EE) and energy intake (EI) in soccer.

### 3.1. Assessment of Energy Expenditure (EE)

Given the varied nature of soccer training, such as on-pitch sessions, strength and conditioning work and individual programmes, it becomes important to fully evaluate the overall EE undertaken by soccer players. In the fullest sense, such an assessment requires measures of resting metabolic rate (RMR), energy cost of meals, the energy needed to undertake various day to day activities, as well as the energy cost of training and matches. The largest overall contribution to EE is training and match play for most soccer players unless resting or injured, when RMR may predominate.

Assessment of a player’s RMR provides a baseline level of EE, which has been found to vary dependent on the age group [33,34]. The normal method for assessment of RMR is through indirect calorimetry in a laboratory-based setting [35]. However, this approach is not always practical within soccer due to the required specialist equipment and time-consuming process to collect the data on a regular basis. Therefore, previous research has attempted to create RMR prediction equations primarily based on height, body mass and reported physical activity level [36]. Such approaches have been criticised due to the fact these are derived from non-athletic populations and fail to account for fat free mass (FFM) [37]. Hannon and colleagues [33] recently developed a novel equation in academy soccer using dual-energy X-ray absorptiometry (DXA) assessments of FFM through stepwise multiple regression which resulted in the following formula:RMR (kcal day^−1^) = 1315 + (11.1 × FFM in kg)

The authors found stronger correlations between the novel FFM-based equation and RMR derived from indirect calorimetry compared to the commonly used equations in sport. Jagim and colleagues [38] examined five well-known predictive equations in a group of 28 athletes. The authors observed that all the prediction equations significantly underestimated RMR, with the Cunningham equation having the best prediction formula and with the lowest error value of 284 kcals. The Cunningham equation is displayed below:RMR (kcal day^−1^) = 500 + 22 × FFM (kg)

It appears that for male soccer players predictive equations can form a basis for estimating RMR when it is not possible to determine RMR by direct laboratory-based measures. Typical daily values range from ~1700–2200 kcal day^−1^ and are mainly dependent on body weight [33].

Estimations of EE in training and match play settings has previously employed measures of oxygen uptake (VO2), heart rate (HR) telemetry and use of doubly labelled water (DLW). The DLW method involves consuming a single bolus dose of hydrogen and oxygen stable isotopes in the form of water based on the individuals body mass. Urine samples are then collected periodically during the assessment period from which the elimination rates of deuterium and 18O are measured via mass spectrometry. Consequently, the difference between the initial sample (after the isotopes have reached equilibrium) and a second sample sometime later, provides an average EE value across the days assessed. Whilst the DLW method is seen as the gold standard for EE assessment, its expense limits practical application to research settings within soccer [39]. Furthermore, DLW realistically is only helpful in providing weekly averages rather than day to day or within-day variations, and most certainly individual training sessions and matches cannot be assessed by this means.

The measurement of VO2 in soccer players could potentially provide information around the relative intensity of soccer-based activities and an estimation of EE [40]. Whilst VO2 can be measured directly in the laboratory using incremental exercise protocols, the lack of ecological validity of such approaches may limited its application. Subsequently, portable gas analysers (e.g., Cosmed k4 system) have been utilised in order to understand the energetic demands during soccer activities directly. It was previously observed that oxygen uptake ranged from 2.5–4.5 L min^−1^ during moderate and high-intensity soccer activities such as a dribbling track [41,42]. Rodríguez and Iglesias [43] measured VO2 using a portable gas analyser during friendly match play, revealing an energy expenditure rate of 11.5 kcal min^−1^. Such systems are sensitive to changes in both speed and shuttle-based running in soccer players [44]. Despite the portable nature of gas analyser systems, they still provide a practical barrier for daily use in soccer players due to both the set-up time required, potential interference with player movement and lack of use in competitive matches where it is not possible to play carrying a device strapped to back/front and wearing a mask.

Assessment of EE through HR telemetry has been widely adopted and is based on the individual relationship between exercise intensities such as walking, jogging, running and so on with laboratory assessed VO2 values [45,46]. The main limitation of using HR is the almost flat slope of the relationship at low expenditure levels [47]. At relative rest, slight movements can increase HR, whilst EE (i.e., VO2) remains almost the same [48]. During intermittent exercise such as soccer, there also appears to be a delay in the HR response when players fluctuate between low to high intensity movements [49]. It appears that the use of HR may be useful for estimates of expenditure for an overall group [50], but not necessarily accurate for individuals. Another practical issue is that players often only wear HR belts during training sessions and not during match-play. Thus, even if HR is adopted to monitor expenditure, large parts of the overall weekly microcycle could be potentially missing without the inclusion of match data.

One proposed solution to the issues with regards to using HR monitoring has been the development of motion sensors using triaxial accelerometers for assessment of daily physical activity [50,51]. Whilst their use to assess EE during free living has been found to be reliable and valid [52], significant underestimation has been found for exercise-based activities [50]. Gastin and colleagues [53] recently compared two commonly used triaxial accelerometers with a portable gas analyser during team sport movements, revealing significant underestimation by up to ~30%. One possible alternative is the use of combined accelerometery and HR with branched-equation modelling which has been previously validated during both free-living and low-to-moderate intensity exercise [54,55,56]. It must be noted that such validation is device-dependent, with some models lacking predictive accuracy at an individual rather than group level [54]. Campbell and colleagues [57] reported poor measurement agreement when comparing such technology against the DLW. There is currently a distinct lack of validation information around the use of triaxial accelerometers and those combined with HR within soccer-specific populations. Therefore, caution must be applied if practitioners are using such devices on a regular basis to quantify EE within their soccer players.

The use of global positioning systems (GPS) is now commonplace within soccer to quantity the external load undertaken by players [58]. The initial use of GPS was to quantify the distances and velocities covered by players in order to differentiate intensities based upon the threshold approach [20]. Common thresholds for such movements include high speed running (>5.5 m s) and sprint distance (>7.0 m s) to separate from lower intensity movements [58]. Osgnach and colleagues [59] further expanded such measurements by estimating the energy cost of movement by also including acceleration-based calculations that were previously neglected. Such calculations, termed ‘metabolic power’ (Pmet), have been widely adopted by practitioners and used to produce common metrics such as average metabolic power, high power (>20 W kg^−1^) and estimated EE [58]. However, when compared to indirect calorimetry, Pmet appears to underestimate EE during team sport movements involving intermittent exercise [60,61]. This may be due to significant underestimation during the recovery phases (i.e., low intensity) when VO2 remains high from the previous exercise but Pmet is low due to a lack of physical movement [62]. Indeed, it has been recently acknowledged that Pmet cannot be used as a measure of EE during team sport activity as originally suggested [63]. The measure is able to quantify the energetic cost of changing speed (i.e., combination of acceleration and speed changes) but fails to detect changes of direction, tackles, jumps, etc. [63]. Therefore, caution must be adopted when using this common measure within the overall EE estimation.

Assessment of total EE as well as EE during soccer training and matches is a useful tool in order to determine the nutritional requirements of players both within a day as well as following a training session/match and across weekly microcycles. This section highlights some of the current benefits and limitations of commonly used methods to estimate EE within practical settings. Practitioners must be aware of the potential errors when interpreting EE data for planning individual nutrition practices.

### 3.2. Assessment of Energy Intake (EI)

There are several dietary analysis methods that have been previously employed within soccer, each technique having positive and negative aspects. The most common method of EI assessment is the use of a recorded food diary, which is typically employed for a period of 3–14 consecutive days in order to gain an insight into a soccer players typical dietary pattern [64]. Using this method, players are instructed to record food intake based on portion sizes from household measurements, such as cups, dishes, and spoons. However, this method is limited in its approach due to the variety in the household items used for comparison both within and between players [65]. In order to overcome this limitation, it has been previously recommended to weigh all food on a set of scales prior to consuming, thus more accurately determining the portion size of each item or meal [66]. Whilst this provides some level of standardisation between players, it has been previously suggested that such recording of intake can influence the usual food choices and thereby alters intake during the recording period [67]. An important issue with employing a weighed food intake method is the length of the dietary recording period. Long duration assessments (>7–14 days) reduce the effect of day-to-day variation, increase bias of reporting and may result in what is termed ‘recording fatigue’ [66]. This notion refers to underreporting of EI due to participants failing to consistently measure and record food intake during the assessment period due to a lack of motivation [66].

The 24-h dietary recall method has also been adopted in soccer, with the same limitations found compared with paper based food diaries [68], although 24-h recall does limit the burden placed on the player. This method requires an experienced practitioner to ensure the correct and adequate information is extracted. Of course, the 24-h recall only provides information on what has been consumed in the previous 24-h period and so is a snapshot of dietary intake. Several 24-h recalls are necessary to determine variations in dietary intake for different days across a typically training and match microcycle.

The recent development of smartphone technology has led to an increase in the available technology used to quantify EI in both general and sporting populations. One common example is the use of phone applications that contain nutritional databases in which the user can input portion size based on either known values or manual entry [69]. Such methods have been found to provide similar accuracy to the previous paper based diary methods, whilst also being more widely accepted as a preferred method of recording [70]. However, when measured in comparison to total EE measured by the DLW method, such applications have been found to significantly underestimate EI [66]. The employment of smartphone technology for dietary intake assessment appears to be an attractive, easy to use alternative for players and practitioners. However, the current validation research is lacking within soccer and therefore caution must be proffered with this approach. Indeed, a recent investigation [71] found that both experienced and inexperienced sport nutritionists underestimated the energy intake from photographs of simple and complex meals up to 18%.

Nevertheless, it is evident that a wide range of techniques are available and advances in technology can assist in the recording of EI in soccer players. A recent investigation [39] chose to include multiple methods to provide a more robust measure of EI and to increase confidence in their in-season reporting with elite soccer players. Self-reporting via 7-day food diaries was the primary method used to collect the data, but this was preceded by an initial dietary habits’ questionnaire or 24-h recall. This qualitative measure was included to establish habitual eating patterns and subsequently allow follow up analysis of food diaries. Technology was included to cross reference the diaries using the remote food photographic method to increase the accuracy of reporting and gain an understanding on portion sizes. A further measure was in place to enhance reliability by cross checking the food diaries with a 24-h recall after one day of entries. This in-depth approach to EI appeared to support the EE, as measured by the DLW method, with no significant difference observed between the daily mean EI and EE, and no changes in body mass over the 7-day period.

So, it appears that a balance needs to be struck to get robust information concerning EI by performing the data collection over a sufficiently long enough period to gather useful data but also limiting the chance of ‘recording fatigue’ or collection of insufficient information. Carrying out the data collection over selected days with gaps between these days has proved useful; notably differentiating between match days, pre-match days, high training load days and recovery/off days.

### 3.3. Energy Expenditure: Training and Match Day

During a typical in-season microcycle in soccer, players will play one competitive match per week which is preceded by several training days with variations in loading. This involves higher loading days (i.e., MD-4 and MD-3) and lower loading days (i.e., MD-2 and MD-1) with respect to number of days away from the next fixture [6]. In addition, starting players will generally have a higher load across a microcycle when compared to non-starters, such as bench players or non-squad players [8]. This can be further exacerbated for elite teams who regularly play 2–3 games per week competing across numerous competitions [7]. Within a microcycle, players will also undertake recovery days post-match (i.e., MD + 1 and MD + 2 for starters) and days off in which the training load will be minimal. Players will also undertake significantly higher loads during the pre-season phase when compared to in-season data [72]. Therefore, it’s important that practitioners are able to gather knowledge around soccer players EE across these different types of sessions and phase across the annual macrocycle.

Early literature exploring the EE practices of professional senior soccer players utilised HR data and estimated EE based upon HR-VO2 regression lines determined during incremental running on a treadmill. Reilly and Thomas [73] conducted initial work using HR measurements during match-play in professional English league players, revealing EE values of ~1510 kcal. Bangsbo [1] reported that EE values during a single match were approximately ~1360 kcal based upon HR data in Danish professional players. In addition, a physiological review of the literature [40] approximated the EE during match play to be between 1519–1772 kcal dependent on the VO2max values of the player (ranging from 60 to 70 mL kg min). Energy Expenditure has also been reported in U20 soccer players, with reports of mean match EE values of 1540 kcal across 12 competitive matches using HR data in outfield players [74]. Based upon the HR-VO2 regression approach, it would appear that competitive match-play accounts for around ~1300–1800 kcal per match based upon aerobic fitness levels and playing position.

When quantifying training EE, previous studies have used different methods including DLW [34,39,75,76,77], accelerometery [78] and estimation equation [79,80,81,82] approaches (Table 1). As eluded to earlier, the DLW method only provides the average EE over the sample period rather than a daily breakdown. In senior soccer players, DLW based data has found that outfield English Premier League players expend 3566 kcal day across a two-game weekly in-season microcycle [39]. Similar data was also found in Japanese professional players (3532 kcal day) across a two-game weekly microcycle using the DLW method [77]. Brinkmans and colleagues [76] collected DLW data across a two-week period during the early phase of the in-season. The authors revealed group average EE data of 3285 kcal day, which included goalkeepers and outfield players. Although there were no positional differences observed for absolute calorie expenditure per day, when expressed relative to overall body mass (BM) and LBM, it was found that midfielders had significantly higher EE compared to goalkeepers (LBM: 50.1 vs. 43.0 LMB day). This data is supported by others [75] who, using a case study approach, revealed daily EE of 2894 kcal day (33.8 kcal kg day) in a Premier League goalkeeper using the DLW method. The observed differences compared to outfield players is likely due to the lower game involvements of a soccer goalkeeper, which is reflected in lower external load data during training and match play [83].

In adolescent soccer players, there is an increased importance to strategically plan nutritional practices in order to ‘fuel’ the demands as the player develops physically and technically. investigations have examined EE in Premier League academy players [34] using the DLW method across a 2-week in-season period. The authors found increases in the overall daily EE of youth soccer players from the U12–U18 age groups in an English Premier League academy (2859 to 3586 kcal day), with similar expenditures observed with the older academy players and their senior peers. The gradual increase in EE was primarily attributed to the increased training load and match demands as players move through the age groups within an academy structure. Conversely, others [78] have found lower daily EE values (2552 kcal day) within an U16 English Premier League academy. This difference in values may be attributed to the methodology used to measure EE i.e., DLW [34] vs. tri-axial accelerometery [78]. Interestingly, the authors also separated the data across a 7-day microcycle period, including training, match and recovery days. It was found that EE was similar on ‘heavy’ double session training days and match days (~2900 kcal). The ‘moderate’ single session training days had an EE of approximately ~2400 kcal, with recovery days EE around ~2150 kcal. However, the lack of training load data from this study limits the contextual interpretation of how the EE relates to the training and match demands during the sample period.

Russell and Pennock [82] also separated EE into training, match and rest days across a weekly in-season microcycle in English U18 Championship academy players. The authors revealed significantly higher EE on match days (~3900 kcal) compared to training days (~3500 kcal). In addition, rest day EE data was significantly lower (~3000 kcal) compared to training days. These observed higher values may be attributed to the use of estimation equations to assess EE, which incorporate BMR, EE during all physical activity and the thermic effect of food. Indeed, other studies within adolescent soccer players utilising estimation equations for EE have revealed higher training and match values compared to the previous studies using the DLW method [79,80,81].

Based upon the current literature on EE observed in senior and adolescent soccer players, there appears to be several limitations and directions for future research. There appears to be a lack of consensus around which method to quantify EE is most valid and practical. This makes comparison of data across studies difficult due to the range of different methods used to quantify EE. In addition, the limitations of certain methods (e.g., DLW method) means that a clear MD- breakdown of EE has yet to be established in modern day soccer players. The lack of data on senior players is also a limiting factor, likely due to teams not being comfortable with 1st team players undertaking research studies alongside their playing duties. Therefore, researchers are encouraged to further develop accurate practical methods of daily EE data collection and develop research relationships with 1st team practitioners in order to allow senior player data collection.

## 4. Nutritional Intake and Soccer

In order to provide evidence-based nutritional guidelines, it is important to first establish the dietary habits of soccer players across different sessions (e.g., training sessions, matches and rest days). This includes identifying energy balance (EI and EE) and macronutrient trends across both different countries and also age groups. The following will discuss the current literature on the dietary habits of both senior and academy soccer players.

Table 1 presents the current available data on the EI and EE habits of senior soccer players. In terms of EI, players consume around ~2200–3000 kcal day during training days [39,76,85]. When factoring in both training and match days combined across the sampling period, this value increases to around ~3100–3900 kcal day [39,75,76,77,84,86,87,92]. On non-training days (i.e., rest days), EI has been reported to be ~2510 kcal day [76]. Jacobs and colleagues [24] also reported high EI of ~4900 kcal day on the day following match-play in Swedish professional players. In the studies that have directed compared energy balance through EI and EE, a negative energy balance of around ~200–500 kcal day across training and match-play combined is observed [39,76,77]. This may be due to players wanting to limit any increases in body fat during the competitive season, which is seen as a negative by both players and coaching staff.

Having identified the link with fatigue and pre match glycogen stores, clear evidence has emerged that indicate a periodised approach to CHO feeding during weekly training in preparation for competition. It is clear from a recent investigation [39] within the English Premiership, albeit with only six players, that CHO intake increases on match day to 6.4 ± 2.2 g·kg^−1^ BM compared with a training day CHO intake of 4.2 ± 1.4 g·kg^−1^ BM. These findings confirm the groups previous work on training and match loads [7], which supports the implementation of a periodised CHO feeding strategy to support physical output and dietary planning. Similar nutritional results, with a much larger sample (n = 41) of professional players from the Dutch Eredivisie [76], support the observation of a periodised CHO approach with a significant increase on match day (5.1 + 1.7 g·kg^−1^) compared to training days (3.9 + 1.7 g·kg^−1^) and rest days (3.7 + 1.7 g·kg^−1^). Such a periodised approach to CHO intake is matched by the overall EI of Premiership players with higher match day EI of 3789 ± 532 kcal compared to training day EI of 2956 ± 374 kcal. Findings reported for the Dutch Eredivisie are 3114 ± 978 kcal, 2637 ± 823 kcal and 2510 ± 740 kcal for match day, training days and rest days respectively.

In the studies reported above, protein did not differ statistically throughout the competitive week in either the English or Dutch players, although a notable observation was the high protein intake observed from the English players with an absolute protein intake of 205 + 30 g; which in relative terms amounts to a daily protein intake of 2.4 g day. Observations from the Dutch players [76] and an earlier finding from two Scottish Premier league teams two decades previously [84] demonstrated lower protein intakes of 129 + 36 g (1.7 g·kg^−1^) for the Dutch players, and 103 + 26 g (1.3 g·kg^−1^) for one team and 108 + 20 g (1.4 g·kg^−1^) for the second. The protein intakes observed with the English players may be considered extreme in the light of the view that intakes in excess of approximately 1.7 g·kg^−1^ are unlikely to aid further muscle building or repair [93], and more recently Morton and colleagues [94] stated that protein intakes at amounts greater than ~1.6 g kg day do not further contribute resistance exercise training-induced gains in fat free mass. These potentially excessive intakes of protein do not represent an issue for CHO intake if players are undertaking a one game week with recommendations ranging 3–8 g·kg^−1^ for CHO [95]. However, greater protein intakes may replace CHO needs that arise during congested fixture periods where CHO is recommended to be 6–8 g·kg^−1^ for CHO [95]. These data provide further incentive to follow a periodised CHO intake and plan in accordance with seasonal fixtures to ensure CHO is not neglected.

For the Academy player, the primary aim of the nutritional strategy is to provide sufficient energy required in order to optimise maturation and physical development and to eventually prepare them for the demands of senior match play [64]. This is particularly important around the phases of peak height velocity (PHV), in which players often have a ‘growth spurt’ that significantly increases overall body mass [96]. As well as EE that was previously shown to increase throughout the academy structure, EI has also been found to increase across the age groups [64,90]. Interestingly, the relative EI per day is actually higher in the younger age groups (63 vs. 44 kcal·kg·day for U12/13 vs. U18 [64]). In terms of energy balance, some previous studies have found a negative balance of around ~100–500 kcal·day [64,79,82], whereas others revealed a positive energy balance during training days [80,81]. Therefore, it may be the case that players are not consuming sufficient calories when combining training and match demands. Being in a positive energy balance is particularly important around PHV, when players are often required to build lean muscle mass during resistance exercise training whilst also expending calories during pitch-based conditioning and team training [97].

In contrast to the senior players, no significant differences in macronutrient intake were observed across a training week that contained four training days, two rest days and a match day [78]. These relatively stable intakes may suggest that academy players do not adjust dietary intake across the week to account for the varying intensity of training and matches. The mean CHO intake was 6.0 + 2.3 g·kg^−1^, 5.6 + 1.6 g·kg^−1^ and 5.5 + 2.0 g·kg^−1^ for heavy training, moderate training and matches respectively. Protein intake followed the same pattern as senior players with a consistent intake throughout the week of 1.6 + 0.5 g·kg^−1^, 1.4 + 0.6 g·kg^−1^ and 1.5 + 0.5 g·kg^−1^ for heavy training, moderate training and matches respectively; albeit these data illustrating an overall lower relative protein intake for academy players. Conclusions made from an investigation into the macronutrient intakes of an English Premiership academy [98], suggested that elite youth soccer players did not meet current CHO guidelines, although this conclusion may be difficult to define as their results are compared to senior guidelines as no such guidelines exist for academy players. Furthermore, there was an apparent difference between academy age specific squads, which was of interest but not understood, with their U18s only consuming 3.2 + 1.3 g·kg-1, much lower compared to the U13/14 (6.0 + 1.2 g·kg^−1^) and U15/16 (4.7 + 1.4 g·kg^−1^). Again, daily protein targets were achieved, although the distribution throughout the day was skewed with great intake of protein at the evening meal, compared to the lowest intake at breakfast.

Mean (95% CI) data from a meta-analysis for EI conducted by Steffl and colleagues [99] identifying research between 2010–2019 illustrates a relatively stable protein intake within the academy and senior players of 1.8 (1.7–1.9) g·kg^−1^ and 1.9 (1.5–2.4) g·kg^−1^ respectively. However, the CHO intake may be of interest as general intakes would appear lower than the recommended amounts in both groups, but particularly for the senior players, 4.3 (3.2–5.4) g·kg^−1^ compared to the adolescences, 5.7 (5.4–5.9) g·kg^−1^. Despite the lower CHO intakes, EI for both groups appears to be acceptable with mean EI between studies observed at for senior players 35.0 (25.5–44.6) kcal·kg^−1^ compared to 41.3 (40.2–42.4) kcal·kg^−1^ for the adolescent players. Therefore, more attention to the CHO intake may be needed especially in the build up to matches and within congested fixture periods where the window of opportunity to fully recover and prepare effectively are limited.

### 4.1. Nutrition for Pre-Match Day (MD-1)

Support for high CHO strategies is routinely suggested for improved performance during simulated soccer activities [92], small-sided games [100] and match play [101], with increased physical output evident throughout. Consequently, the primary nutritional focus for pre-match day, or matchday -1 (MD-1), is to ensure that both muscle and liver glycogen stores are optimally elevated in preparation for matchday endeavours. In terms of training, MD-1 usually focuses on light tactical work and brief activity at high intensity to maintain ‘sharpness’. Nutritionally, the emphasis is for an increase in CHO intake such that 6–8 g·kg^−1^ is consumed [95]. It is possible to optimise muscle glycogen stores in a 24-h period by following a high CHO diet of 10 g·kg^−1^ [102], although this investigation was undertaken with endurance athletes who remained inactive during the study.

It is well recognised that eccentric activities (such as involved in soccer) result in a reduced capacity to replenish muscle glycogen stores [14,24,103,104]. Muscle glycogen stores have been observed to have been recovered by around 80% following a soccer match in elite players after 3 days of high CHO intake [14], and this confirms the 50% reduction in muscle glycogen 48-h after a competitive match [24]. It is important therefore, to ensure adequate CHO intake occurs on MD-1 (hopefully with a reduced training load) to maximise muscle glycogen stores.

In order to achieve a CHO intake of 6 g·kg^−1^ in MD-1 it is necessary to consume CHO at all opportunities. Table 2 illustrates what may be undertaken in order to attain such a goal for a 75 kg player, where the CHO intake is likely to be 450 g i.e., 6 g·kg^−1^. It is useful to consider a 4 or 5 meal strategy for MD-1 i.e., breakfast, lunch, mid-afternoon snack, dinner, and supper (the mid-afternoon or supper may not be suitable for all). With such a strategy it is quite possible to ingest 2 g·kg^−1^ CHO at lunch and at dinner, around 1 g·kg^−1^ for breakfast; the remaining CHO consumption (i.e., around 1 g·kg^−1^) may be obtained via appropriate drinks and/or snacks in the afternoon or for supper. Since glycogen is stored with water, fluid consumption with each meal/snack is strongly advised.

Is there any evidence that the variety of CHO consumed with regards to the glycaemic index (GI) is an important consideration? Since high glycaemic index (HGI) foods are digested and absorbed more rapidly than low glycaemic index (LGI) alternatives, it is pertinent to consider consuming foods which have a high GI and may encourage greater intake. However, there is a lack of evidence in regard to whether high or low GI meals are advantageous for glycogen loading over a 24-h period. Burke, Collier and Hargreaves [105] concluded that a HGI nutritional strategy should be followed as part of a recovery strategy following glycogen depleting exercise, with an approximate 50% difference between the variety of GI, but this investigation employed a training volume and intensity unlikely to be followed during MD-1 where the training load is normally light and therefore may not be applicable. More research is required in relation to GI carbohydrate intake during the MD-1. Having said that, it is prudent to allow players to consume carbohydrate sources they enjoy during MD-1 as long as there are sufficient amounts to achieve −6 g·kg^−1^.

### 4.2. Nutrition and Match-Day: Pre-Match

As highlighted above, MD-1 is vital for nutritional preparation for competition and may even be more important than any meals prior to the match on the match day itself. This is likely since it is not possible to replenish significant loss of muscle glycogen within a short period such as 24-h. Certainly, liver glycogen may be more rapidly replenished in a contracted period but not muscle glycogen. If an appropriate feeding strategy of high CHO is adhered to, match day itself should just be a ‘top up’ of muscle and replenishment of liver glycogen. Liver glycogen has been observed to decrease significantly following an overnight fast [106], with total depletion expected after 36 h. This means that approximately 33% of liver glycogen could be lost during an overnight fast of 12 h, and up to 50% if no CHO is consumed until kick off.

If muscle and liver glycogen is to be enhanced on match day, it is pertinent to examine the findings of Wee and colleagues [107] who observed a 15% increase in muscle glycogen 3-h after ingestion of 2.5 g·kg^−1^ of a high GI breakfast but not from a low GI breakfast meal. Unfortunately, the authors did not assess liver glycogen storage. It is possible that liver glycogen stores would be promoted in both conditions since an earlier study [108] using MRS demonstrated significant liver glycogen restoration after exercise when fed 1 g·kg^−1^ glucose or sucrose. In fact, sucrose ingestion resulted in a much higher amount of storage than glucose probably because sucrose contains fructose which has been established to enhance liver rather than muscle glycogen [109].

An issue over the last number of years is that kick off time varies considerably in elite soccer matches (mainly due to the demands of television). Start times may be as early as 12:00 or 12:30 and as late as 20:00, although most matches in the UK have a 15:00 kick off time. Clearly early kick off times imply a single feed strategy whereas for a 15:00 start there is the possibility of a two-feed strategy, and for a 20:00 kick off a three-feed strategy is desirable. For the earlier start times the importance of a sufficient CHO intake during MD-1 is paramount, whilst marginally less so for evening matches. Figure 1 highlights the possible scenarios for match day feeding strategies based on variations in start times.

The general consensus for pre-match nutrition would suggest a relatively high CHO meal of between 1–3 g·kg^−1^ [95] or 1–4 g·kg^−1^ [21] approximately 3–4 h before kick-off. Low glycaemic index CHO meals are thought to be more advantageous than HGI CHO prior to exercise mainly due the possibility that high GI meals could result in so-called rebound hypoglycaemia after about 3–4-h. The application of GI for sport and pre-exercise feeding was first studied in the early 1990 s [110] and concluded that LGI foods produced a greater endurance capacity compared to HGI foods, partly due to a potential glycogen sparing mechanism attributable to a more stable insulin and glucose profile. Further investigations supported this notion that LGI CHO foods improved capacity [111] and time trial performance [112], although these investigations were based around continuous exercise models. In contrast, soccer specific investigations illustrated no difference between HGI and LGI pre-match foods [113] or meals [114], with the latter investigation following a more ecological protocol of match day timings, prior breakfast, and realistic meals with all macronutrients present (details of meals presented in Table 3). Even though significant differences in postprandial glucose responses were observed, no metabolic or performance difference were observed [114]. Subsequently, muscle glycogen measurements were analysed following a further soccer specific simulation to investigation glycogen utilisation differences between GI, and although a significant difference was observed between the control trial (fasted) and both HGI and LGI, no significant difference was seen between the fed trials [115].

The concept of a high fat meal has also been investigated and compared to an isocaloric LGI meal prior to a soccer specific simulation [116]. Recommendations for fat intake are not specified for soccer players with the same rigor as CHO and protein. Although a key macronutrient to provide energy and an important source of fat-soluble vitamins and essential fatty acids (α-linolenic acid and linoleic acid), fats are usually reduced to allow addition CHO and protein. A recent meta-analysis [99] identified a recent trend of fat intake of ≥30%, but no lower than the recommended lower limit of 20%. Results into the pre-exercise high fat meal identified no performance benefits between this and the LGI meal when muscle glycogen was adequate [116]. Differences were observed between fat metabolites, substrate oxidation rates and subjective ratings of fullness and hunger, yet no transferable effect on performance or subjective rating of perceived exertion. It is important to note that prior to the soccer specific investigations [114,116], participants followed their habitual diet with a greater focus on CHO, signifying normal glycogen stores. Therefore, these results may advocate flexibility and a more personalised approach in the pre-match meal, rather than a prescribed meal or static menu, if a normal high CHO diet has been maintained throughout the training week. These may suit players as personal preferences can be maintained due to foods choices, cultural difference/choices, habits, rituals and/or superstitions.

Hydration is an important consideration since dehydrations is a well-recognised fatigue factor in prolonged activities, and water is required for glycogen storage. Consequently, players need to ensure they start the match in a euhydrated state. Evidence based on assessment of urinary parameters demonstrates that a large proportion of players may start competition in a relatively dehydrated state [93]. Therefore, to ensure players are well prepared with regards to their fluid status, it is recommended that they slowly imbibe between 5–7 mL·kg^−1^ at least 4-h prior to the match [117]. From a practical perspective, the consumption of water is advisable during and after the pre-match meal up until arrival at the match venue. Use of carbohydrate-containing beverages should be discouraged after the meal and only be re-introduced after warm-up and in the 5–10-min period before the match starts. This is to reduce the likelihood of rebound hypoglycaemia ensuing in the first 10–15 min of the match.

An inevitable consequence of undertaking any form of exercise such as soccer (training or match play) is that muscle protein breakdown (MPB) ensues, and that muscle protein synthesis (MPS) is attenuated [118]. It is also well established that feeding protein in the hour prior to engaging in such bouts of activity and in the 1 or 2 h afterwards results in an increase in net protein balance whereas no protein intake causes a deficit in net protein balance [119]. Consuming a pre-match meal with some protein 3-h before is helpful, although it may be advisable to consider drinking some form of whey protein (without carbohydrate) an hour or so before kick-off. Such a proposal may help reduce MPB during the match, although no scientific data has been presented to date. The ingestion of a protein bar or protein shake is a useful consideration here.

One of the few supplements that has strong evidence for improving soccer performance is caffeine. Acute caffeine intake of a moderate dose of caffeine (3–6 mg·kg^−1^) before exercise has the capacity to improve several soccer-related abilities and skills such as vertical jump height, repeated sprint ability, running distances during a game and passing accuracy [120]. Data suggest that caffeine improves the physical and technical elements of performance that are required for successful soccer match play. The ingestion of 2–6 mg·kg BM^−1^ of caffeine has been reported to increase repeated sprint and jump performance [121], reactive agility [122], jump height [123] and passing accuracy [124] during intermittent exercise protocols, replicating the physical demands of soccer.

The information presented above is illustrated in Figure 2, with the nutritional intake planned before kick-off, ensuring optimal glycogen stores and hydration from the start of competition.

### 4.3. Nutrition and Match-Day: During Match

In order to maintain sufficient, hydration, muscle glycogen and blood glucose concentrations, sufficient intake of CHO and fluid are the main nutritional considerations during match-play. Data from laboratory-based studies show that CHO ingestion during soccer-specific exercise augments plasma glucose availability and maintains rates of CHO oxidation [125]. However, given the acyclic nature of activity in soccer, there are no scheduled breaks where fluid can be consumed; besides, gastric tolerance and the perception of gut fullness do not allow for suitable rehydration for soccer players. Furthermore, it has been demonstrated that the intensity corresponding to that of a soccer match is sufficient to slow gastric emptying [128]. Therefore, due to the continuous nature of play, with infrequent, unscheduled stoppages, the only two occasions that a player is guaranteed to be able to consume fluid are before the game and at half-time. It is therefore prudent that players take on board fluid during and additional breaks in play e.g., injury stoppages.

The intensity of exercise associated with a competitive match is high enough to induce appreciable heat load, causing players to lose up to 3 L of sweat during a match [31]. Therefore, as a guide, players should aim to drink sufficient fluids to prevent a deficit of no more than 2–3% of pre-match euhydrated body mass [117,129]. The addition of CHO to this fluid can further improve exercise capacity [130], possibly due to prevention of hypoglycaemia, maintenance of high CHO oxidation rates, glycogen sparing, and effects on the central nervous system are important considerations, delaying the onset of fatigue [131]. Consequently, a CHO intake at a rate of 30–60 g·h^−1^ has been associated with a consistent beneficial effect on performance in soccer [132]. However, players in the English Premier League reported a CHO intake of 32 ± 22 g·h^−1^ just before and during a match [39].

The ingestion of CHO has frequently resulted in improvements in exercise capacity during the performance of exercise protocols that simulate the work-rate of soccer [25,120,130] and actual match-play [120,133,134], although sprint performance is less consistent. However, soccer performance is not only dependent on physical performance. Motor skills and cognitive performance also play a crucial role and there is a tendency for players’ skills and cognitive performance to decline during the latter stages of a match [135]. Carbohydrate has been shown to attenuate, or even eliminate, this detrimental effect over the course of a match [136]. For example, Harper and colleagues [137] reported that a CHO–electrolyte solution improved dribbling speed during the later stages of a soccer match simulation. Furthermore, CHO supplementation attenuated the decrements in shooting [138] and passing [136] performance during simulated soccer match-play. Furthermore, the co-ingestion of ingestion of a 6% CHO solution with the addition of 160 mg·L^−1^ of caffeine has been reported to improve sprinting performance and countermovement jumping [121], in addition to reductions in perceived exertion following a normal high CHO feeding strategy with the co-ingestion of 5 mg·kg^−1^ of caffeine prior to a simulated soccer protocol [139]. A further interesting development is that caffeinated gum containing 200 mg of caffeine (2 × 100 mg pieces of gum) prior to soccer-specific tests has been reported to enhance aerobic capacity countermovement jumping performance by approximately 2%, although further research is warranted [140].

It should also be noted that matches can extend to extra time and penalty shoot-outs. As previously mentioned [32], it was reported that CHO–electrolyte gel ingested before a simulated extra-time period raised blood glucose concentrations and improved dribbling performance during the extra-time period of simulated soccer match-play. Finally, CHO mouth rinse has been shown to increase self-selected jogging speed with likely benefits for repeated 15 m sprint performance [141]. Therefore, CHO mouth rinsing prior to extra-time or penalty shootout could potentially enhance performance in situations where CHO consumption is limited by gastrointestinal concerns.

### 4.4. Match-Day: Post-Match

Due to the depletion of muscle and liver glycogen during match play the main focus for recovery is to replenish these stores in addition to rehydrating and ensuring muscle protein is recovered. Normally, this should be achieved within a 2-h window after a match [142] and is especially important if fixtures are close together i.e., within 2–3 days or so. This limited time frame for glycogen resynthesis is due to exercise-induced glycogen depletion promoting activation of glycogen synthase [143], exercise-induced increases in insulin sensitivity [144], and exercise sensitisation of muscle cell membranes to glucose delivery [127]. Therefore, the sooner the CHO intake after exercise the better. Those players who are unable to consume adequate CHO following a match risk a reduced ability to resynthesis glycogen by approximately 50% throughout a 4-h period, regardless of consuming CHO 2-h following the cessation of exercise [142]. In the same fashion as MD-1 when the nutritional goal is to maximise glycogen synthesis, HGI CHO are the variety of choice to enhance the replenishment over the LGI equivalent [105], with an ideal intake of approximately 1.0–1.2 g·kg·h^−1^ to achieve this [127]. If sufficient CHO cannot be ingested in the immediate post-match period, there is some evidence that additional protein may help glycogen resynthesis. Thereby, if the recommended intake of 1.2 g·kg·h^−1^ is unable to be attained, then an additional amount of 0.4 g·kg·h^−1^ of protein could ensure adequate rates of glycogen storage [145].

In addition to the use of protein to support muscle glycogen replenishment, it is a vital component of nutritional recovery due to the muscle protein breakdown that will occur during match play. Previous understanding suggested that 20 g of protein provided no greater advantage over 40 g during the recovery period following resistance exercise [146,147]. However, this was observed following a bout of unilateral exercise and not full body exercise, which may be more comparable to football. MacNaughton and colleagues [148] thereby conducted a similar investigation following whole body resistance exercise and found that 40 g stimulates a greater myofibrillar protein synthesis response than 20 g in young resistance-trained men, irrespective of their lean body mass. Solid or whole foods rich is protein would generally be recommended, but whey protein shakes are often favoured as a more expedient way to consume protein, with appetite sometimes suppressed following exercise. Whey protein is recommended over casein and soy protein due to its rapid absorption, and importantly it’s high proportion of leucine [149], which is a key amino acid that at high doses can stimulate muscle protein synthesis.

Another important aspect of recovery from match play is rehydration. Players can be hyperthermic at the final whistle and sweat losses can exceed 3 to 4 L [93]. Therefore, the goal post match is to replace any fluid and electrolyte deficit. As players should be consuming regular foods during recovery that contain sodium, plain water should suffice [117]. However, if sodium is not present or insufficient, this will delay a return to a euhydrated state due to the stimulation of urine production, and it may be recommended to intake formulated sports drinks for their added electrolyte content. As an approximate recommended intake to achieve rehydration quickly, 150% of weight loss or 1.5 L per 1 kg of weight lost may be suggested [128], which would support the fluid loss and account for any urination that ensues. Furthermore, it is key not to hyperhydrate post-match, especially following evening matches, as sleep disturbances in athletes has been attributed to waking throughout the night to urinate [150]. Therefore, milk may be a drink of choice prior to bedtime, not only for the protein and casein content, but due to its high electrolyte content compared to water [151].

Due to the nature of the game at professional levels, many substitutes are available for selection at various times. These players are required to load up on fluid and carbohydrates in the same manner as those starting the match. However, if they play only a few minutes or not at all, then it is strongly advisable not to engage in the high CHO eating strategies outlined above. Rather, these players should focus on low carbohydrate, high protein foods both in the changing room and for their post-match meal. Failure to do so over repetitive matches is likely to lead to increases in body fat due to the conversion of the excess CHO ingested into fat. Clearly not an advisable situation! Table 4 below provides some suggestions for foods and drink to consume post-match.

### 4.5. Day after Match Day (MD + 1)

A day that is sometimes forgotten, or at least not treated with the same nutritional respect is MD + 1. Yet nutritional importance is paramount as players may still be in an energy deficit and need to consume appropriate CHO to continue their refuelling strategy in preparation for the forthcoming training week and match/es. Early work conducted by Jacobs and colleagues [23] with elite Swedish players highlighted that insufficient CHO intake due to poor dietary habits following a competitive fixture limited post-match glycogen replenishment. They observed values that were typically lower than sedentary individuals 48 h post-match. Furthermore, the importance of MD + 1 is emphasised further during periods of fixture congestion, which demonstrate increased volume and intensity of activity due to match play [39]. Therefore, a continuation of high CHO intake is advised on MD + 1 since this will continue to top up muscle glycogen as well as liver glycogen stores.

However, recent investigations [152] have shown that glycogen resynthesis may not be equal in both fibre types, with an impairment observed in type II fibres 48 h post simulated match play. This was despite players following a high CHO diet supplemented with whey protein, strengthening the rationale for the continuation of high CHO for complete recovery. The muscle damaging effects of eccentric exercise, characterised in soccer by dynamic changes of direction and high intensity accelerations and decelerations, are suggested to cause the impairment in glycogen resynthesis seen in type II fibres [104]. Muscle damage can reduce the translocation of GLUT4 [153], obstructing glucose uptake into the muscle cells [154], delaying the replenishment process. Therefore, in order to overcome these impairments, soccer players are advised to maintain a high CHO diet during the first 24 h following competition. For this purpose, it is suggested that around 6–8 g·kg^−1^ BM is continued [95]. Supplementing the CHO diet with protein has already been discussed [145] and can be advantageous. Further nutritional strategies to support replenishment could be supported with the addition of creatine. Research suggests that 20 g·day of creatine monohydrate may provide further provision to increase muscle glycogen alongside a high CHO diet [155]. Improvements in glycogen replenishment were observed 24 h post-exercise and maintained throughout 6 days of feeding, reinforcing creatines inclusion to further augment muscle glycogen storage to use as part of the recovery strategy.

## 5. Conclusions

In light of the evidence presented on match day and recovery, Figure 2 showcases the nutritional needs and timings around this competition day, capturing all the important timepoints to consume food or drink prior to the match, during the match, and for immediate post-match recovery.

However, the dietary habits of a soccer player need to meet the demands of the training and preparation for matchday as well as a full recovery. Research highlighted in the current review suggests that soccer players consume appropriate amounts of protein, but the periodisation of CHO intake may still be flawed, or recommendations are too excessive. Consequently, the need for optimal nutritional intake throughout the competitive season is imperative for peak performance. The communication between the multidisciplinary team within a soccer club is important to ensure that players are fuelling correctly for each training session, the build up to competition, and not forgetting total recovery.

## Figures and Tables

**Figure 1 nutrients-14-00657-f001:**
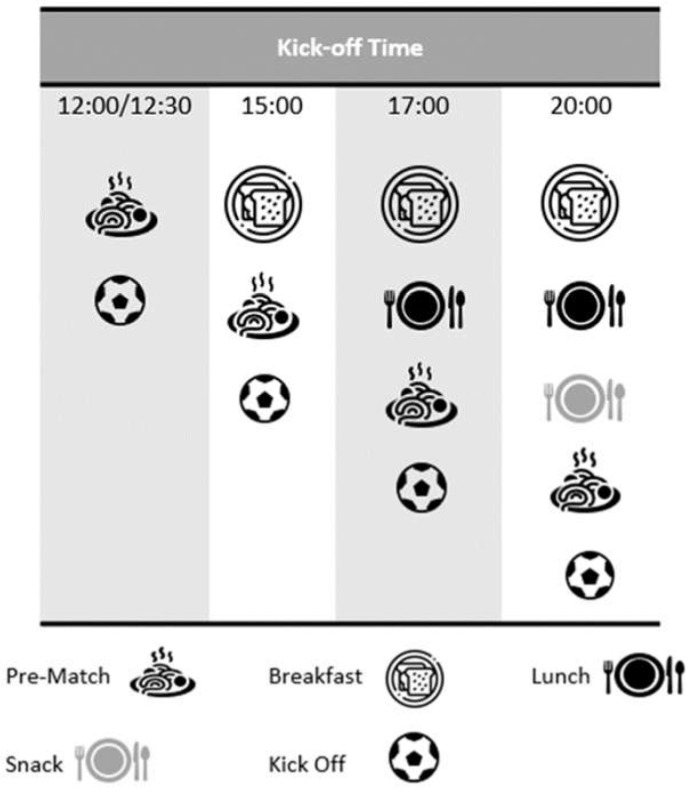
Pre match feeding strategies based on kick-off times.

**Figure 2 nutrients-14-00657-f002:**
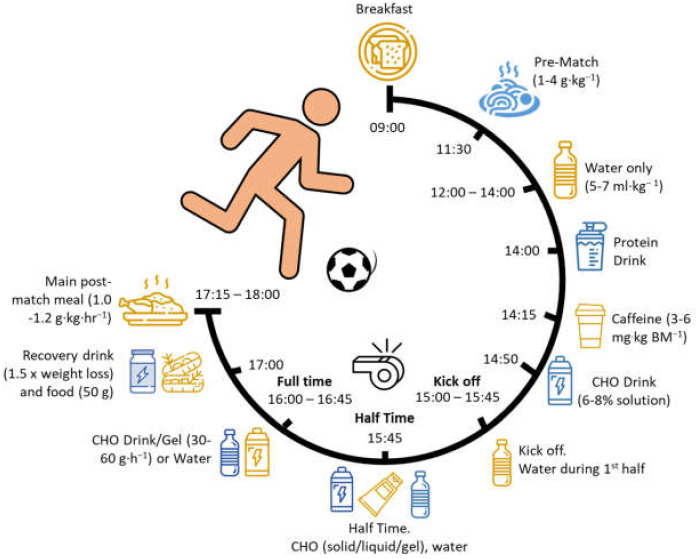
Schematic of a typical nutrition scenario for a player undertaking a soccer match starting at 15:00 (requirements based on [21,117,120,125,126,127]).

**Table 1 nutrients-14-00657-t001:** Energy intake and expenditure of male soccer players during training and match play.

					Energy Intake		Energy Expenditure	
Reference	Study Population	Age/BM	Period	Method	kcal·day	kcal·kg·day	kcal·day	kcal·kg·day
**Senior Soccer Players**								
Anderson et al. [39]	EPL Professional Players (*n* = 6)	27 ± 3 years 80.5 ± 8.7 kg	7-day In-Season	In—Food Diary Ex—DLW	T = 2956 ± 374 M = 3789 ± 532	T = 36.7 M = 47.1	3566 ± 585 (overall T + M)	44.2 (overall T + M)
Anderson et al. [75]	EPL Goalkeeper (*n* = 1)	27 years 85.6 kg	7-day In-Season	In—Food Diary Ex—DLW	3160 ± 381 (overall T + M)	36.9 (overall T + M)	2894 (overall T + M)	33.8 (overall T + M)
Bangsbo et al. [84]	Danish Professional Players (*n* = 7)	20–28 years 62.7–85.9 kg	3-day In-Season	In—Food Diary Ex—n/a	3749 (overall T + M)	49.2 (overall T + M)	-	-
Bettonviel et al. [85]	Dutch Eredivsie Professional Players (*n* = 29)	20 ± 4 years 73 ± 8 kg	4-day In-Season	In—24 h Recall Ex—n/a	2988 ± 583 (overall T + M)	40.9 (overall T + M)	-	-
Brinkmans et al. [75]	Dutch Eredivsie Professional Players (*n* = 41)	23 ± 4 years 77.6 ± 8.0 kg	14-day In-Season	In—24 h Recall Ex—DLW	T = 2637 ± 823 M = 3114 ± 978 R = 2510 ± 740	T = 33.9 ± 10.6 M = 40.1 ± 12.6 R = 32.3 ± 9.5	3285 ± 354 (overall T + M)	42.4 ± 3.5 (overall T + M)
Devlin et al. [83]	Australian Professional Players (*n* = 18)	27 ± 5 years 75.6 ± 5.6 kg	1-day Pre-Season	In—24 h Recall Ex—n/a	T = 2247 ± 550	T = 29.7	-	-
do Prado et al. [86]	Brazilian Professional Players (*n* = 118)	23 ± 1 years GK (*n* = 12) 83.9 kg CD (*n* = 20) 83.9 kg WD (*n* = 21) 69.7 kg MID (*n* = 41) 70.8 kg ST (*n* = 24) 72.1 kg	Habitual Food Inquiry	In—Interviews Ex—n/a	GK: 3903 CD: 2961 WD: 3361 MID: 2989 ST: 3641 (overall T + M)	-	-	-
Ebine et al. [76]	Japanese Professional Players (*n* = 7)	22 ± 2 years 69.8 ± 4.7 kg	7-day In-Season	In—Food Diary Ex—DLW	3113 ± 581 (overall T + M)	44.6 (overall T + M)	3532 ± 408 (overall T + M)	50.6 (overall T + M)
Jacobs et al. [24]	Swedish Professional Players (*n* = 15)	20–30 years 68–92 kg	3-day In-Season	In—Food Diary Ex—n/a	4947 ± 1126 (recovery post-M)	67.3 (recovery post-M)	-	-
Maughan [87]	Scottish Professional Players (*n* = 51)	Team A = 26 ± 4 years 80.1 ± 7.8 kg Team B = 23 ± 4 years 74.6 ± 6.5 kg	7-day In-Season	In—Food Diary Ex—n/a	Team A = 2629 ± 621 Team B = 3059 ± 526 (overall T + M)	Team A = 32.8 Team B = 41.0 (overall T + M)	-	-
Ono et al. [88]	EPL and League One Players (*n* = 24)	n/a	4-day In-Season	In—Food Diary Ex—n/a	2648–4606 (period n/a)	-	-	-
**Adolescent Soccer Players**								
Briggs et al. [78]	EPL Academy Players (*n* = 10)	15 ± 0 years 57.8 ± 7.8 kg	7-day In-Season	In—Food Diary Ex—ACC	2245 ± 321 (overall T + M)	38.8 (overall T + M)	2552 ± 245 (overall T + M)	44.2 (overall T + M)
Caccialanza et al. [89]	Italian Serie A Academy Players (*n* = 43)	16 ± 1 years 69.8 ± 7.4 kg	4-day In-Season	In—Food Diary Ex—n/a	T = 2560 ± 636	T = 37.2	-	-
Ersoy et al. [79]	Turkish Academy Players (*n* = 26)	16 ± 1 years 67.3 ± 5.9 kg	3-day Pre-Season	In—Food Diary Ex—EQ	T = 3225 ± 692	T = 47.9	T = 3322 ± 240	T = 49.4
Hannon et al. [34]	EPL U12/13 Academy (*n* = 8)	12 ± 0 years 43.0 ± 4.8 kg	14-day In-Season	In—Photo Ex—DLW	2659 ± 187 (overall T + M)	63.0 ± 8.0 (overall T + M)	2859 ± 265 (overall T + M)	66.5 (overall T + M)
Hannon et al. [34]	EPL U15 Academy (*n* = 8)	15 ± 0 years 56.8 ± 6.2 kg	14-day In-Season	In—Photo Ex—DLW	2821 ± 338 (overall T + M)	50.0 ± 7.0 (overall T + M)	3029 ± 262 (overall T + M)	53.3 (overall T + M)
Hannon et al. [34]	EPL U18 Academy (*n* = 8)	18 ± 0 years 73.1 ± 8.1	14-day In-Season	In—Photo Ex—DLW	3180 ± 279 (overall T + M)	44.0 ± 7.0 (overall T + M)	3586 ± 487 (overall T + M)	49.1 (overall T + M)
Iglesias-Gutiérrez et al. [80]	Spanish Academy Players (*n* = 33)	14–16 years 65.1 kg	6-day In-Season	In—Food Diary Ex—EQ	T = 3003	T = 46.5	T = 2983	T = 45.8
Naughton et al. [90]	EPL U13/14 Academy (*n* = 21)	13 ± 1 years 44.7 ± 7.2 kg	7-day In-Season	In—Food Diary Ex—n/a	T = 1903 ± 432	T = 43.1 ± 10.3	-	-
Naughton et al. [90]	EPL U15/16 Academy (*n* = 25)	14 ± 1 years 60.4 ± 8.1 kg	7-day In-Season	In—Food Diary Ex—n/a	T = 1927 ± 317	T = 32.6 ± 7.9	-	-
Naughton et al. [90]	EPL U18 Academy (*n* = 13)	16 ± 1 years 70.6 ± 7.6 kg	7-day In-Season	In—Food Diary Ex—n/a	T = 1958 ± 390	T = 28.1 ± 6.8	-	-
Rico-Sanz et al. [81]	Puerto Rican Olympic Team (*n* = 8)	17 ± 2 years 63.4 ± 3.1 kg	12-day In-Season	In—Food Diary Ex—EQ	T = 3952 ± 1071	T = 62 ± 12	T = 3833 ± 571	T = 60.5
Ruiz et al. [91]	Basque Club Players (*n* = 81)	Team A = 14 ± 0 years 62.8 ± 2.2 kg Team B = 15 ± 0 years 66.7 ± 2.3 kg Team C = 17 ± 0 years 73.6 ± 0.8 kg Team D = 21 ± 0 years 72.9 ± 1.2 kg	3-day In-Season	In—Food Diary Ex—n/a	Team A = T = 3456 ± 309 Team B = T = 3418 ± 182 Team C = T = 3478 ± 223 Team D = T = 3030 ± 141	Team A = T = 54.6 ± 5.5 Team B = T = 51.5 ± 2.5 Team C = T = 48.4 ± 2.4 Team D = T = 41.1 ± 2.1	-	-
Russell and Pennock [82]	English Championship Academy (*n* = 10)	17 ± 1 yrs 67.5 ± 1.8 kg	7-day In-Season	In—Food Diary Ex—EQ	2831 ± 164 (overall T + M)	41.9 (overall T + M)	3618 ± 61 (overall T + M)	53.6 (overall T + M)

Abbreviations: ACC = Accelerometery Methods; CD = Central defender; DLW = Doubly Labelled Water; EPL = English Premier League; EQ = Estimation equations; Ex = Expenditure method; GK = Goalkeeper; In = Intake method; M = Match; MID = Midfielder; Photo = Remote photographic method; R = Rest day; ST = Striker; T = Training; WD = Wide defender.

**Table 2 nutrients-14-00657-t002:** Potential meal ideas to achieve 6 g·kg for a 75 kg player on MD-1. Data analysed with nutritional management software Nutritics (Dublin, Ireland).

*Meal*	*Food Source*	*Amount*	*Amount of CHO*
*Breakfast* *Total—88.1 g* *(1.2 g kg CHO)*	Cereal—Weetabix (with milk)	37 g (135 mL)	31.6 g
	Toast—2 slices (with flora light)	60 g (14 g)	31.0 g
	Fruit cocktail (in juice)	100 g	11.7 g
	Fresh Orange (glass)	160 mL	14.1 g
	Poached eggs × 2	100 g	0 g
*Lunch* *Total—156.6 g* *(2.1 g kg CHO)*	Rice	160 g	50.4 g
	Sweet and sour chicken	160 g	9.4 g
	Broccoli	85 g	3 g
	Green beans	60 g	2.3 g
	Apple crumble & Custard	150 g 100 g	68.4 g
	Fresh Apple juice (tall tumbler)	300 mL	23.1 g
*Dinner* *Total—152 g* *(2 g kg CHO)*	Mashed potato	300 g	41.4 g
	Salmon (white wine sauce)	210 g (121 g)	6.4 g
	Carrots	90 g	5.2 g
	Broccoli	85 g	3 g
	Peas	80 g	7.4 g
	Strawberry (1 cup) & meringue (×2)	160 g/32 g	39.1 g
	Ice cream	35 g	7.4 g
	CHO-electrolyte drink	500 mL	31.5 g
*Drinks/Snacks* *Total—72.3 g (0.96 g kg CHO)*	Slice of fruit cake/loaf	77 g	40.8 g
	CHO-electrolyte drink	500 mL	31.5 g
*Total CHO Intake*	459 g (6.1 g·kg^−1^ CHO)		

**Table 3 nutrients-14-00657-t003:** Details of the ‘pre-match’ meals provided during investigations by Hulton et al. [114,116].

High Glycaemic Index Meal (GI = 80) [114]					CHO (g)	Protein (g)	Fat (g)	Energy (kcal)
Akash rice (63 g)	Chicken Breast (100 g)	Tomato based sauce (300 g)	Lucozade Original (380 mL)	Water (210 mL)	138.8	35.7	23	870.3
Low Glycaemic Index Meal (GI = 44) [114]					CHO (g)	Protein (g)	Fat (g)	Energy (kcal)
Brown basmati rice (63 g)	Chicken Breast (100 g)	Tomato based sauce (300 g)	Apple Juice (590 mL)		133.7	37.9	23.7	866.3
High Fat Meal [116]					CHO (g)	Protein (g)	Fat (g)	Energy (kcal)
Egg fried rice (75 g)	Chicken breast (100 g)	Korma sauce (200 g)	Milkshake (200 mL)	Double cream (50 mL)	59.4	35.3	70.2	995.6

**Table 4 nutrients-14-00657-t004:** Some ideas for nutritional support in the changing room after a match and for a post-match meal.

	CHANGING ROOM		POST-MATCH MEAL	
	Player	Substitute	Player	Substitute
**FLUIDS** 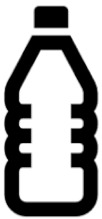	1 L CHO-electrolyte OR 1 L CHO-protein shake	300 mL protein shake (no CHO) Water	Fresh fruit juice	Water
**MAIN MEAL** 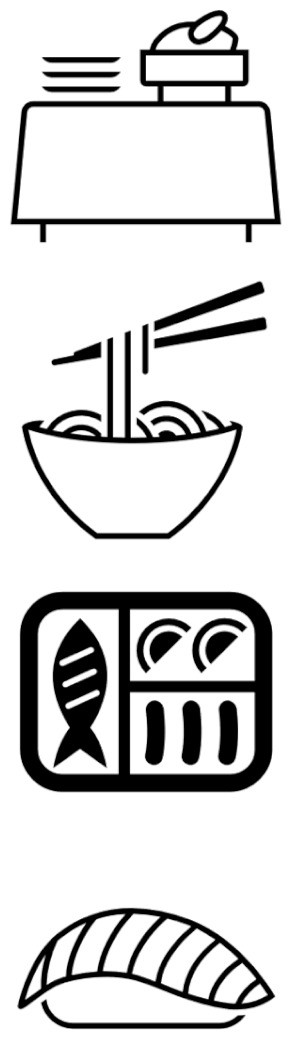	Baked wedges Pizza slices Sushi Chicken goujons & dip Prawn goujons & dip Sliced frittata	Chicken goujons + dip Prawn goujons & dip Sliced omelette Chicken Kebab	Pasta meal Curry, rice & naan Sweet & sour meal Chicken kebab & rice Paella Cottage pie Salmon/cod/tuna steak, vegetables & mashed potato Frittata & fries Jerk chicken, rice & peas	Chicken or beef salad Prawn stir fry Chicken kebab with salad Bolognaise & Courgetti Omelette & beans Roast meat & vegetables Salmon/cod/tuna steak & roasted vegetables
**DESSERT** 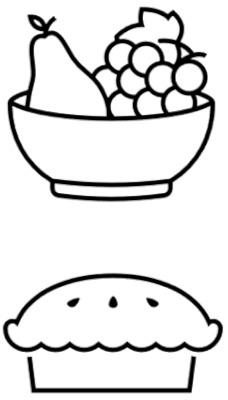	Meringue Fresh pineapple slices	Apple slices	Sticky toffee pudding Banoffee pie Fruit crumble Eton mess	No Dessert

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
