# Peer review of "Energy Requirements and Nutritional Strategies for Male Soccer Players: A Review and Suggestions for Practice"

_nutrients, 2022, doi:10.3390/nu14030657_

Round 1

Reviewer 1 Report

Review

Nutrients-1534612

Dear Authors,

I have received your manuscript entitled “Energy requirements and nutritional strategies for male soccer players: A narrative review and suggestions for practice”, which is a review of the literature on the energy requirements and needs related to the high physical effort of male soccer players, as well as, the related nutritional requirements that improve training and thus, above all, their game during matches.

These two important factors, energy expenditure during training and match, as well as proper nutrition, which have a positive effect on performance and proper play of mentioned in study male soccer players, were described in accordance with the aim of the study. After analysing the collected literature were described it, not only in the narrative form but also in tables and figures, and in conclusions, which can be directly used by interested trainers.

Minor comments according to editorial form (I hope will be improved) and used units:

For example, once you write kcal/day, another time you write kcal.day-1 (I think you should write this dot as superscript as another in line 414) or kcal/day in all places also in tables.  I have marked in pdf version (a lot of this,  please decide one style).

 E.g. line 348 "EE of 2894 kcal/day (33.8 kcal.kg/day) ..." are you sure? I think so that have to be 33.8 kcal/kg/day.

E.g. line 414 "... CHO intake increases on match day to 6.4 +/- 2.2 g / kg ..." please explain per kg of what? athlete's body weight? or per kg of food delivered? Please check all in pdf.

You are not explaining some of the abbreviations that I also highlighted in the pdf.

Moreover, in line 806 in conclusions, I see “Figure 4” (Is this a link to an appendix ?), but in manuscript is only Figure 1 and 2.

I hope you will find the comments helpful for the improvement of the manuscript.

Sincerely Yours,

Reviewer

Author Response

Dear reviewer, 

Thank you for your valuable time to review our narrative review, we appreciate it is quite a large manuscript. We thank your summary of our paper and for highlighting the errors and lack of consistency in our reported units.

We have improved these throughout the paper using a consistent method. Thanks for highlighting!

With regards to your comment:

line 414 "... CHO intake increases on match day to 6.4 +/- 2.2 g / kg ..." please explain per kg of what? athlete's body weight? or per kg of food delivered? Please check all in pdf. 

We do have BM after the kg, but your have rightly indicated that we have not defined what BM is previously. Therefore I have added 'body mass (BM)' on line 346 as this is the first time we use body mass.

We have not defined VO2max further as we feel this is a know measure and any explanation would affect the flow of the paragraph. For the equations used on lines 156 and 163, the abbreviations have been included in the text. previously. 

Many thanks.

Reviewer 2 Report

Energy requirements and nutritional strategies for male soccer players: A narrative review and suggestions for practice.

A well-written review that addresses the energy requirements of football for all categories. It also makes recommendations for meeting these nutritional requirements.
Commentary: A reference to the contribution of fats to the energy requirements of football and a suggestion on the types of fats to be consumed would be useful.
Furthermore, apart from salmon, references to fish consumption are missing. Finally, in my understanding, there should also be a recommendation for the consumption of olive oil. 

Minor comments:

pp 806: wrong figure's number (probably the number is 2)

pp 810: In Figure 2, it would be useful to record the recommended amounts of the food or drink (gr/kg).

pp 811: Information about the numbers, in the parentheses, must be reported in the legend of figure 2

Author Response

Dear reviewer, 

Thank you for your valuable time to review our narrative review, we appreciate it is quite a large manuscript. We thank your summary of our paper and for highlighting some errors and suggestions to improve this.

We have added a section on fat intake after we initially mention the pre-exercise high fat meal (line 609). We have added some additional fish alternatives within table 4 so it reads 'Salmon/cod/tuna steak, vegetables & mashed potato' for more variety. We have not included anything on olive oil. We appreciate that it is a healthy fat, and something that might be added in the cooking process but we haven't gone into that specific detail. We haven't included any real detailed information on supplements that may include fish oils either. 

Thank you for identifying our mistake with the number of figure 2, that has been amended. In addition, we have also included some references that would support the numbers in the parenthesis.

many thanks.